# Using TanDEM-X Global DEM to Map Coastal Flooding Exposure under Sea-Level Rise: Application to Guinea-Bissau

**Morto Baiém Fandé [1], Cristina Ponte Lira [2,3,*] and Gil Penha-Lopes [1]**

1   Centre for Ecology, Evolution and Environmental Change (cE3c), Faculty of Sciences, University of Lisbon, Edifício C2, Piso 5, Campo Grande, 1749-016 Lisboa, Portugal; mbfande@fc.ul.pt (M.B.F.); gppenha-lopes@fc.ul.pt (G.P.-L.)
2   Instituto Dom Luiz (IDL), Faculdade de Ciências, Universidade de Lisboa, Edifício C1, Piso 1, Campo Grande, 1749-016 Lisboa, Portugal
3   Departamento de Geologia, Faculdade de Ciências, Universidade de Lisboa, Edifício C6, Piso 3, Campo Grande, 1749-016 Lisboa, Portugal
*   Correspondence: fclira@fc.ul.pt

**Abstract:** The increased exposure to coastal flooding in low-lying coastal areas is one of the consequences of sea-level rise (SLR) induced by climate changes. The coastal zone of Guinea-Bissau contains significant areas of low elevation and is home to most of the population and economic activity, making it already vulnerable to coastal flooding, especially during spring tides and storm surges (SS). Coastal flooding will tend to intensify with the expected SLR in the coming decades. This study aimed at quantifying and mapping the area exposed to the coastal flooding hazard using SLR scenarios by the years 2041, 2083, and 2100. The study analyzes and discusses the application of a the simple "bathtub" model coupled with a high-precision global digital elevation models (TanDEM-X DEM) to areas where no other data are available. Therefore, three coastal hazards hot-spots of Guinea-Bissau: Bissau, Bubaque, and Suzana, were used as case study. At each site, the area potentially exposed to coastal flooding was evaluated in a geographic information systems (GIS) environment, by estimating the Total Water Levels for each SLR scenario. For all areas, land exposed to coastal flooding hazard increases significantly and progressively with increasing SLR scenarios. Bissau and Suzana, where housing, infrastructure, and agricultural land are low-lying, presented the greatest flood exposure, while Bubaque, where housing and infrastructure are located in relatively high-lying land and rain-fed agriculture is practiced, present lesser flood exposure. The methodology presented is simple to use but powerful in identifying potentially vulnerable places to coastal flooding hazard, and its results can aid low developed countries to assess their exposure to coastal risks, thus supporting risk awareness and mitigation measures.

**Keywords:** total water level; "bathtub" model; geographic information systems

## 1. Introduction

Flooding is one of the most common, expensive, and dangerous hazards that can affect coastal areas. They include episodic flooding, caused by temporary extreme conditions (e.g., storm surge, tsunamis, inland flooding, high-tide flooding) and long-term flooding by a rise in sea-levels (locally or globally).

One of the consequences of sea-level rise (SLR), induced by the increasing global average temperature, is the added risk of coastal flooding in low-lying coastal areas [1–6]. The IPCC Fifth Assessment Report stated that global mean sea-level (GMSL) rose by an average of 0.19 m in the last century (1901 to 2010) and estimates an increase up to 0.98 m by 2100. Recently published studies suggested even higher increase rates during this century [7–9]. Additionally, other studies indicate that even if climate is stabilized by reducing greenhouse gases (GHG) emissions, GMSL will continue to rise after 2100 due to past emissions [10,11].

As sea-level rises, the vulnerability and risk to flooding on coastlines will increase. This fact is even more worrisome since it is projected that by 2060, 12% of the world's population will live in coastal zones with elevation <10 m above mean sea-level (MSL) [12]. Negative impacts of coastal flooding due to SLR include loss of human life and agricultural production, damage to housing and infrastructure, as well as human displacement [3,10,13,14]. These impacts are expected to be particularly severe in least developed countries (LDC), given the already existing vulnerability, together with a rapid population growth and increased occupation of low-lying coastal zones for livelihood [12].

Although LDC are particularly vulnerable, there is a deficiency in coastal research studies due to a lack of data and knowledge. In the specific case of coastal flooding impacts, good topographic data are essential, but are often absent in these countries [15,16], thus, researchers will have to rely on spaceborne global (G) Digital Elevation Models (DEMs) [17]. GDEMs present higher uncertainties when compared with high-resolution datasets, but naturally cover less surveyed areas. Most GDEMs are freely available open-source products [18], but their accuracy may vary [16,17,19].

TanDEM-X is an Earth observation radar mission that provides global DEMs at two different resolutions: (1) a non-freely available version at 0.4 arc-second (~12 m), but available exclusively for scientific purpose through the Science Phase application [20]; and (2) a free-to-download global DEM—TanDEM-X 90—at 3 arc seconds resolution (~90 m) [17,21]. According to Rizzoli et al. [19] TanDEM-X dataset presents: (1) relative height accuracy presents a 90% confidence level of 2 m and 4 m for flat and steep terrain, respectively), which is met on a global scale for 97.76% of all geocells not disclaimed due to volume decorrelation effects; (2) more than 50% of all DEM geocells and about 70% of them show a relative height accuracy at 90% confidence level better than 1 m for flat terrain and better than 2 m for steep terrain; (3) at global scale, the relative height accuracy at 90% confidence level for vegetated areas remains under 2 m for both flat and steep terrain. Hawker et al. [17] provide a comparison between TanDEM-X 90 and other popular global DEMs: Shuttle Radar Topography Mission (SRTM) DEM and the error-reduced version of SRTM called Multi-Error-Removed-Improved-Terrain (MERIT) DEM [22]. The authors specifically focus on low slope floodplains, as: flood inundation modelling is predominantly sensitive to height errors [17,23], concluding that the average vertical accuracy of TanDEM-X 90 and MERIT are similar, being both a significant improvement on SRTM.

The present study aims at presenting an assessment of coastal flooding exposure under sea-level rise at one of the LDC countries where a lack of detail information and increased vulnerability exist—Guinea-Bissau. Guinea-Bissau is characterized by a very-low topography, high concentration of human population, infrastructures, and economic activities (agriculture, tourism, fishing, etc.) near/on the coastline and is home to rich ecosystems and wetlands [24,25]. The population density in coastal areas <25 km inland grew from 9 inhabitants/km$^2$ in 1950 to 60 inhabitants/km$^2$ in 2008 [26]. De Sherbinin et al. [27] estimated that in 2010 Guinea-Bissau's coastal zone <10 m height was home to ~135 k people and, in 2050, the number is expected to grow more than five times (~737 k people). Guinea-Bissau's coastal areas are already affected by coastal flooding immediate impacts [25,28], with crop loss due to coastal flooding as one of the main risk factors contributing to the lack of availability or access to food [29]. This risk is likely to increase under the expected rapid SLR, thus coastal flooding assessment in different SLR scenarios is necessary to provide accurate information in support of decision-making on coastal planning, adaptation, and resilience. However, such works are scarce for Guinea-Bissau. Fandé et al. [28] give a first estimation of coastal flooding extension for Bissau city, considering SLR scenarios for year 2100. In addition to this first estimation, and to our knowledge, no authors have yet addressed the mapping and quantification of coastal flooding at the local level in Guinea-Bissau using SLR scenarios.

The present study intends to present the first coastal flood extension for SLR scenarios at three hotspots of coastal hazards of Guinea-Bissau (Bissau city, Bubaque Island, and Suzana Section) using a simplified methodology—single surface model [30–32] and global

Digital Elevation Models. The chosen study areas correspond to sections where coastal flooding has already been identified or is likely to occur: Bissau city has documented coastal flooding events [28], Suzana section and Bubaque island have been pin-pointed as coastal erosion hot-spots [33,34]. SLR scenarios considered years 2041, 2083, and 2100 following the projections of [7] for Conakry city (Republic of the Guinea) and the highest emissions scenarios—RCP 8.5 (see Section 3.3). A fourth scenario for 2100 considers an extreme value, where the collapse of the Antarctic ice sheet becomes more likely.

## 2. Study Areas

This work selected three low-lying coastal sections, which represent typical coastal environments of Guinea-Bissau: (1) Bissau City (hereinafter referred to as Bissau) is the country capital, located in the North Center; (2) Bubaque Island (hereafter referred to as Bubaque) is located in the South and part of the Bijagós Archipelago; and (3) Suzana Section (henceforth referred to as Suzana) located at the Northwest (Figure 1).

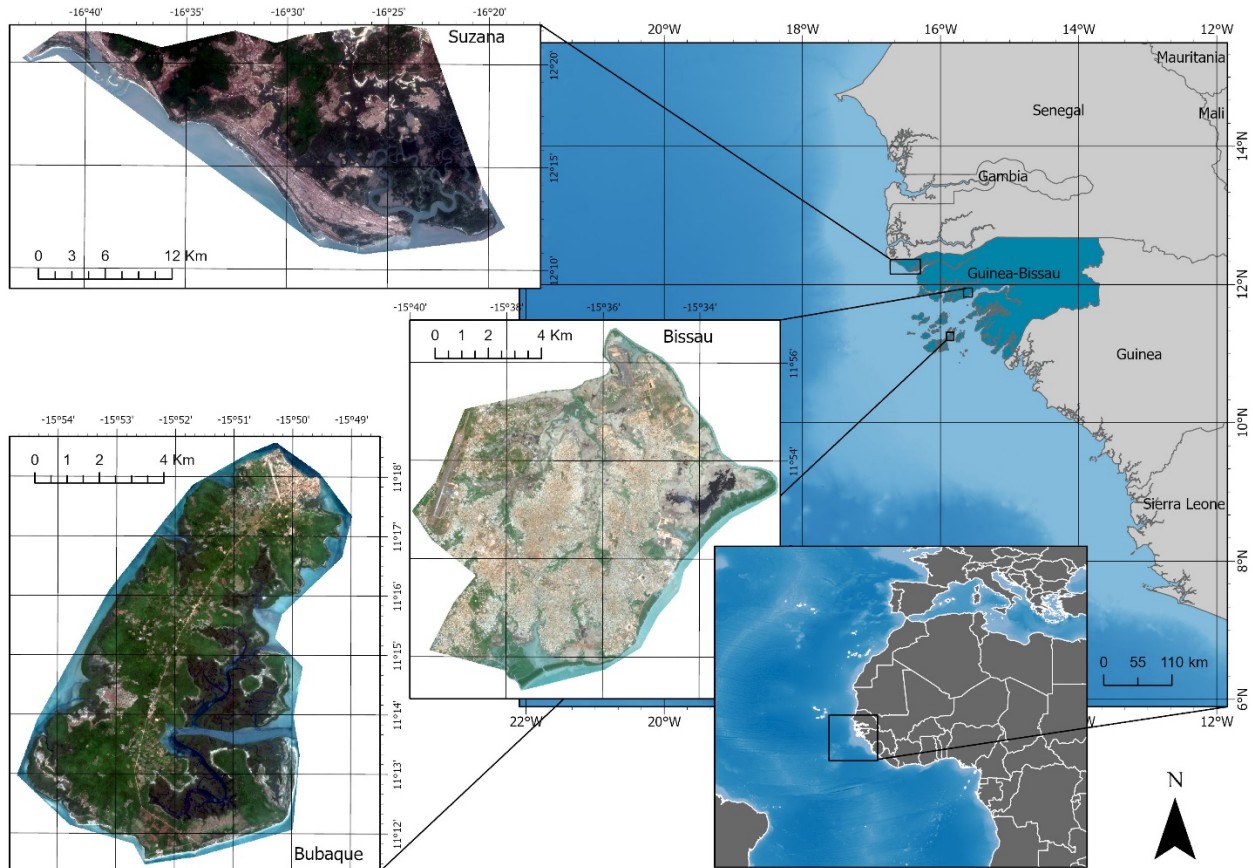

**Figure 1.** Location map of the three study areas, Bissau, Bubaque, and Suzana. Mesoscale view of Guinea-Bissau and neighbor countries coastline also shows the bathymetry in shades of blue (darker blue deeper water, light blue shallow waters). Bathymetric information from ArcGIS Online. Map Coordinates: WGS1984 UTM Zone 28 N.

The delimitations of Bissau and Suzana do not correspond to official administrative limits: For Bissau, a polygon was created covering the peripheral urban areas in rapid expansion and whose residents share the same services and way of living, officially defined as Bissau Autonomous Sector. In this way, the area at risk of flooding is extended using urban and occupation criterion to the detriment of the exclusive use of administrative boundaries. For Suzana, due to the lack of official maps with delimitation at Section administrative level, the limitation with the São Domingos Section was carried out through

a polygon with fictitious limits and bordering around the villages belonging to the Suzana administration Section. For Bubaque Island, the natural limits of the island were considered.

The three areas present great differences in terms of socioeconomic and environmental characteristics (Table 1). The most populated area is Bissau, an urban area having as main economic activities trading and services. Suzana and Bubaque have roughly the same population and are both more rural than the urban Bissau, with main economic activities of agriculture and fishing, whereas Bissau's main activities are services and trade. Bubaque is also a touristic area. In terms of geomorphology, Bissau is located on the river Geba estuary, with floodplains, swamps, and mangrove. Suzana coastline is open to the Atlantic, and exhibits sandy beaches, dunes and cliffs, and small river mouths, with the presence of some mangrove. Bubaque island presents a coastline that alternates between sandy beaches, mangroves, and cliffs.

**Table 1.** Socioeconomic and environmental characterization of the three study areas, Bissau, Bubaque, and Suzana. MSL—mean sea-level.

| Characteristics | Study Area | | |
| --- | --- | --- | --- |
| | Bissau | Bubaque | Suzana |
| Total area (excluding water bodies) (km$^2$) | 94.90 | 63.03 | 319.67 |
| Area below 5 m above MSL (%) | 28.85 | 24.08 | 32.82 |
| Area below 10 m above MSL (%) | 41.23 | 40.90 | 65.23 |
| Maximum elevation (m) above MSL | 58.00 | 50.05 | 45.00 |
| Coastline (km) Average annual rainfall (mm) * | 13.07 1250–1750 | 44.78 1750–2000 | 51.30 1250–1500 |
| Population ** | 365,097 | 6427 | 6701 |
| Main economic activities | Services, trade | Agriculture, fishing, trade, tourism, craft | Agriculture, fishing |
| Land occupation | Urban | Urban/rural | Rural |
| Coast type | River Geba Estuary, sandy/slimy floodplain, swamps, mangrove, ports | Island, sandy beaches, cliffs of yellow-reddish sand, mangrove | Mostly open to the Atlantic Ocean and without mangrove, small rivers with mangrove, dunes and sandy beaches, cliffs of reddish-yellow sand, barrier islands |

* Average values for the 1981–2010 period [35]; ** Data from the last census in Guinea-Bissau, carried out in 2009 [36].

## 3. Materials and Methods

To assess the extent of coastal flooding, the methodology adopted followed the single-surface or "bathtub" model [30–32]. This simple approach uses only two variables: a single inundation value and terrain elevation information, which allows representation of a planar water surface of the predicted inundation level.

### 3.1. Single Inundation Value—TWL Approach

To estimate the inundation value a Total Water Level (TWL) approach was used, where TWL sums up all vertical components that contribute to water level [37]. TWL is governed by the precautionary principal, thus all components correspond to the maximum observed or estimated. TWL is represented by the following water vertical components: astronomical tide (AT), storm surge (SS), and sea-level rise (SLR), represented by the expression:

$$TWL = AT + SS + SLR \qquad (1)$$

### 3.1.1. Astronomical Tide

Astronomical tide (AT) is the periodic variation in sea water levels due to the attraction exerted by the Sun and the Moon on the Earth (among other components), with strictly known periodicities [38]. In this study, the astronomical tide was determined with reference to the maximum of the equinoctial high tide for year 2015 and obtained from the expression:

$$AT = MaxT - HZ, \tag{2}$$

where MaxT is the maximum high tide, and HZ is the hydrographic zero.

AT values referenced to the MSL can be observed on Table 2. Estimations of MaxT were obtained using WTides Software version 3.1.12 (www.wtides.com (accessed on 1 June 2020), Warkworth, New Zealand), which uses harmonic data to obtain tide height, and data from local ports closed to the study areas. For Bissau, data from port of Bissau were used; for Bubaque, data referring to the port of Bubaque; and for Suzana, where there are no local records, the study assumed the average of Caió and Djogue ports, which are located on the open coast facing the Atlantic and close to Suzana coastline (see Table 2).

**Table 2.** Astronomical tide (AT) estimated for the three study areas. AT of Suzana was obtained from the average data of Caió and Djogue ports. HZ—Hydrographic zero; MSL—Mean Sea-Level.

|  | Bissau | Bubaque | Caió | Djogue | Suzana |
|---|---|---|---|---|---|
| MaxT (m—HZ) | 5.84 | 4.93 | 3.64 | 1.9 | - |
| HZ (m) | 2.89 | 2.54 | 1.9 | 1.02 | - |
| AT (m—MSL) | 2.95 | 2.39 | 2.74 | 0.88 | 1.81 |

Assuming that SLR will have little effect on the height of AT in the 21st century [39], the values obtained were used for all periods.

### 3.1.2. Storm Surge

Storm Surge (SS) is the "abnormal fluctuation of water level resulting from severe atmospheric disturbances, such as strong winds and air pressure changes (usually referred to as typhoons, extratropical cyclones, and other disastrous weather systems), which causes the tide level within the affected area to far exceed the usual level" [40] (p. 440). Its determination is essential to map coastal flooding risk areas.

The only quantitative data available on SS for the region are data from the Global Disaster Alerting Coordination System concerning Tropical Storm Fred, that occurred in 2015. Values range from 0.2 to 0.3 m for Guinea-Bissau coast [41]. In this study, the value of 0.3 m was adopted, by applying the precautionary principle. Given the persistent uncertainty and lack of scientific consensus on the future storm events intensity and the consequent height of the tide level [14], the value of 0.3 m is used for all periods.

### 3.1.3. Sea-Level Rise Scenarios

SLR will not be uniform in space and time, due to the influence of several natural and anthropic factors at regional or local level [7,42]. Thus, in any particular region or locality, it is the relative sea-level (not the global value) that determines the long-term susceptibility to coastal flooding [43]. This study, due to the lack of local data for Guinea-Bissau, considered the relative SLR projections proposed by [7] for the Conakry city (Republic of the Guinea). Conakry city projections were considered adequate, because: (1) it has a contiguous coastline to Guinea-Bissau; (2) is the nearest location with available projections and (3) both countries continental shelve is very similar in width and bathymetry [44]. Jevrejeva et al. [7] projections are based on the highest emissions scenarios among the four families of the Representative Concentration Pathways (RCP 8.5), because the global historical $CO_2$ emissions (main GHG) have followed this scenario [45]. Furthermore, the high emissions scenario is justified because $CO_2$ emissions continued to increase, even after the recent largest climate agreement (Paris Agreement 2015) to stabilize global GHG

emissions [46–48]. Therefore, this study considers three scenarios: 0.34 m by 2041; 1.22 m by 2083; and 1.95 m by 2100.

Additionally, an extreme sea-level scenario of 5 m for year 2100 was considered, based on the evidence that global mean sea-level may rise more rapidly due to the potential collapse of West Antarctic Ice Sheet [49–52]. The likelihood of such a collapse is low, but it has great potential to cause destructive impacts and its consideration, in flood assessments, is important to encourage policymakers to think about more catastrophic possible occurrences [53], thus allowing to deal with uncertainties and contributing to prevention and long-term sustainability [54]. The worst-case approach has been used in many studies of impacts and adaptation to SLR, e.g., it was applied in Australia [55]; in Ghana [56]; in Great Britain [57], on a global scale [58].

### 3.1.4. Total Water Level

The estimated TWL for different periods and study areas is shown in Table 3. For Bissau the TWL values, for years 2041, 2083, 2100, and 2100 extreme scenarios, vary between 3.59 m for the less conservative scenario and 8.25 m for the extreme one; for Bubaque between 3.03 m and 7.69 m; and for Suzana between 2.45 m and 7.11 m. It is also apparent, from TWL estimations, that the extreme scenario of 2100 will increase the TWL value in almost 5 m in relation to the less extreme scenario of 2100.

**Table 3.** TWL estimated for different SLR scenarios in each study area, Bissau, Bubaque, and Suzana.

| SLR Scenarios (Year) | TWL (m) | | |
|---|---|---|---|
| | Bissau | Bubaque | Suzana |
| 2041 | 3.59 | 3.03 | 2.45 |
| 2083 | 4.47 | 3.91 | 3.33 |
| 2100 | 5.20 | 4.64 | 4.06 |
| 2100 * | 8.25 | 7.69 | 7.11 |

* Extreme scenario, considering potential collapse of West Antarctic Ice Sheet.

### 3.2. Terrain Elevation Data

Topographic information is essential for coastal flooding mapping, but unfortunately Guinea-Bissau does not have a high-resolution DEM available. Therefore, terrain elevation was obtained from two GDEMs provided by the TanDEM-X Earth observation mission. To have access to the TanDEM-X at 0.4 arc-second (~12 m) and, thus, selecting the most accurate representation of terrain to improve the accuracy of flood predictions and minimize height errors, a scientific proposal was submitted to the Science Phase application of 2016 [20]. The proposal was accepted and included access to DEM tiles which covered the three study areas. Nevertheless, due to data restrictions along the border of Senegal, no tile was available for the Suzana area, hence, this area was covered by a TanDEM-X 90 DEM tile.

### 3.2.1. TanDEM-X Datasets

The TanDEM-X dataset is a Digital Surface Models (DSM), which means that the measured Earth's surface height includes any surface objects, thus not representing the bare surface (or Digital Terrain Model (DTM)) [17]. Although, DSM are not the ideal datasets for flooding studies, they are the only available topographic dataset for several areas of the world, and a common typology of spaceborne global digital elevation models. Wessel [59] states that TanDEM-X 0.4 arc-second DEM presents a mean error of −0.17 m and RMSE of 1.29 m for kinematic GPS points and a mean error of 0.12 m and RMSE of 1.42 m for GPS-on-BM points; RMSE values for ground level in developed and forest areas show slightly higher deviations of ±1.4 m and ±1.8 m, respectively, whereas the low vegetation class remains with a RMSE of ±1.1 m. According to Hawker et al. [17], TanDEM-X 90 DEM, which is an average of the TanDEM-X 0.4 arc-second DEM, presents the best vertical accuracy results at floodplains for bare soil, shrubland, sparse vegetation, and urban land cover types, only surpassed by the MERIT dataset for short vegetation and tree coverage

types. The same authors concluded that TanDEM-X 90 and MERIT have a similar accuracy, and both MERIT and TanDEM-X 90 are more accurate than SRTM, being the latter the most used DEM. Thus, TanDEM-X 90 presents a mean error of 1.06 m, mean absolute error of 1.74 m and RMSE of 3.16 m for floodplains [17]. Both TanDEM-X datasets present the best compromise between availability of data for the study area and accuracy. This is corroborated by Figure 2, where a comparison between the MERIT and the TanDEM-X 90 m DEMs for the area of Suzana is presented. Generally, can be observed that both DEMs are similar, with a $R^2$ correlation of 0.8, but TanDEM-X presents higher values for higher areas and MERIT presents higher values for lower areas. Because both seem to correlate well and the original TanDEM-X dataset is more recent than MERIT, thus representing more truly coastal areas that are in constant change (e.g., sand spits), the TanDEM-X 90 m dataset was the one chosen to use for the Suzana section area.

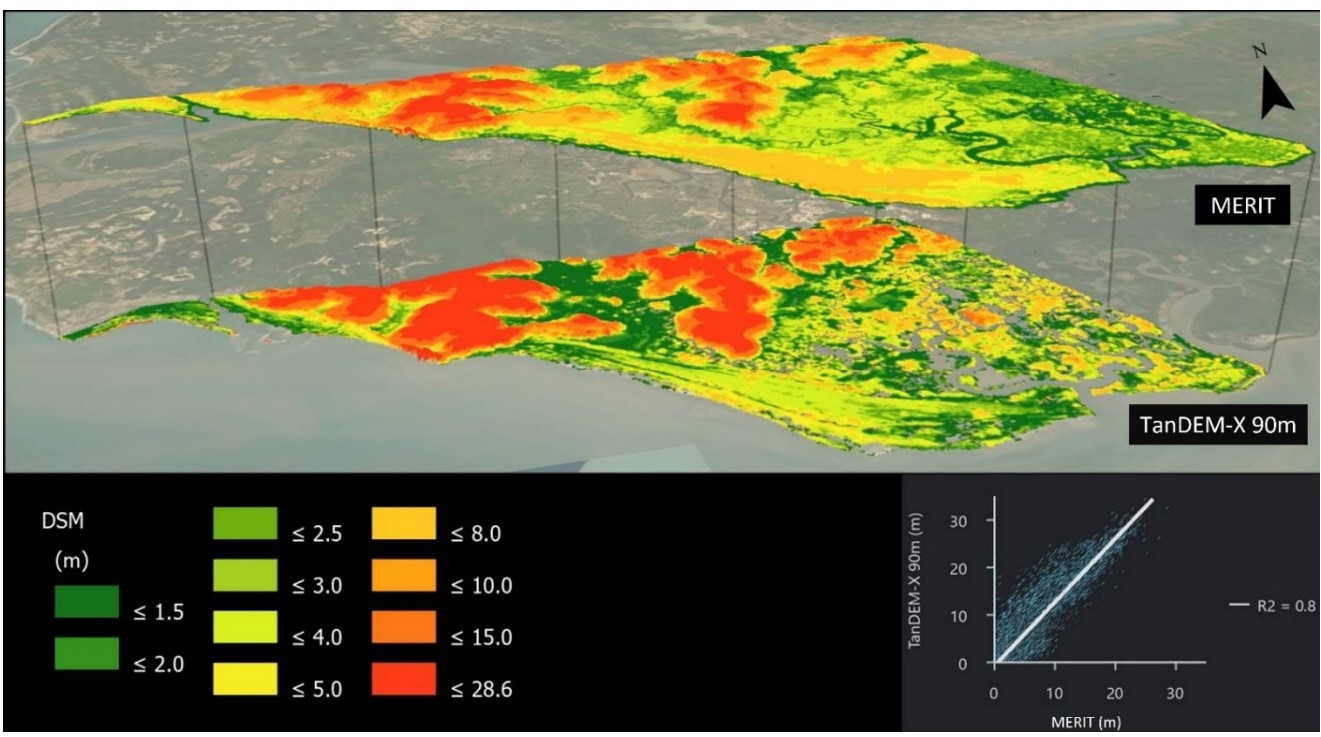

**Figure 2.** Comparison of MERIT and TanDEM-X 90 m DEM in a 3D view for the area of Suzana. Basemap is a Sentinel 2 image.

### 3.2.2. TanDEM-X Processing

Both TanDEM-X DSMs are served by the German Aerospace Center (DLR) as ellipsoidal height, having been converted to orthometric height using the geoid model of the Earth Gravitational Model 2008—EGM08 [60], with a cell size of 2.5 min. This transformation followed the expression:

$$H = h - N, \tag{3}$$

where H is the orthometric height, h is the ellipsoidal height and N corresponds to the model of the geoid used. After this transformation, both datasets were converted to a common datum: the projected coordinate system WGS 84/UTM zone 28N.

The pre-processing steps (Figure 3) of the DEM required the cleaning of anomalous values, located on surfaces covered by water, using a mask of the existing water bodies. For this, images acquired in the near infrared region (NIR—band 8) of the Sentinel-2 satellite (freely available at https://scihub.copernicus.eu/ (accessed on 1 June 2020), EU) were used to produce water bodies mask and so obtaining the final cleaned DEM. S2 image acquired in 29 December 2016, covering the region of Suzana, and in 25 April 2017, covering the other areas, were the best available and recent images to map the water bodies of the study

area. The choice of an adequate threshold value allowed to separate land zones from water. NIR images are often used to delineate water bodies, taking advantage of the fact that this type of radiation is totally absorbed by water, with no reflection back to the sensor (water is represented in dark tones, contrasting with the surrounding land areas).

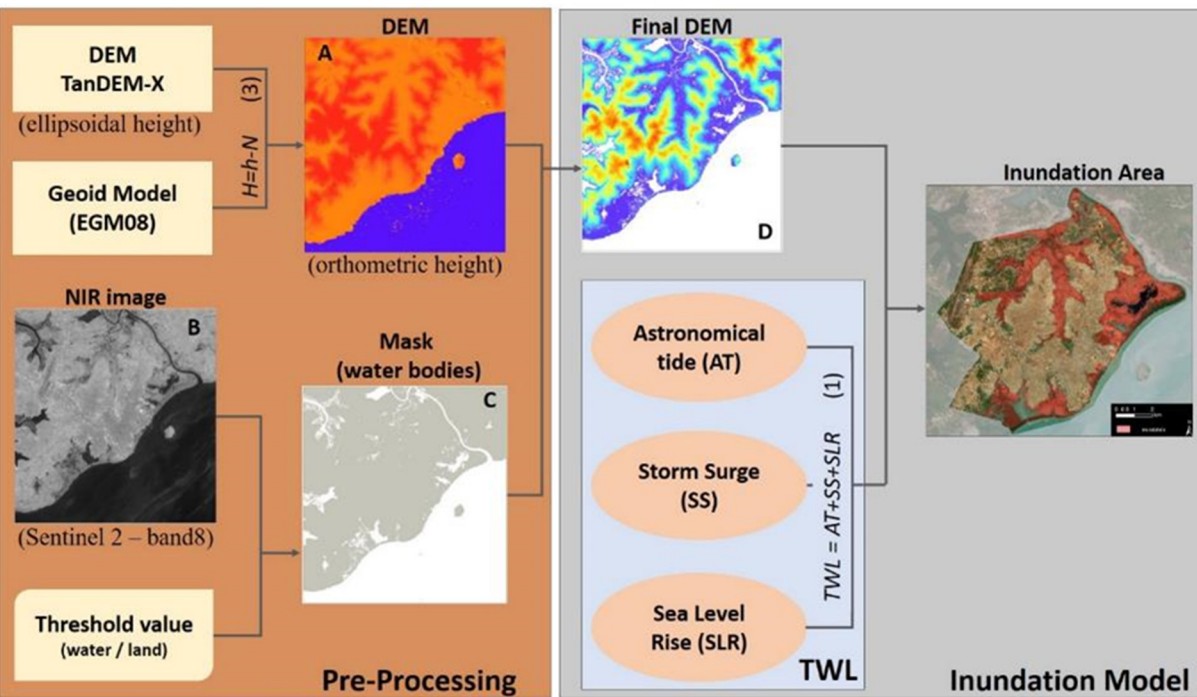

**Figure 3.** Flowchart of the DEM pre-processing procedures and the inundation model used in this study.

For Suzana area, and because TanDEM-X 90 DEM presents a spatial resolution of 90 m, a resample was made to increase the resolution and meet the one of TanDEM-X 0.4 arc- second DEM (12 m). The bilinear interpolation resample technique was used in the resampling procedure, which calculates the value of each raster pixel by averaging (weighted for distance) the value of the neighboring four pixels.

### 3.2.3. Flood Mapping

In this study, it was considered "at risk of flooding" any portion of dry land adjacent to the coastline and with hydrological connectivity to the sea that presents an elevation less than or equal to the value of the considered TWL. Thus, after the classification area affected by each estimated TWL, the cleaning of isolated pixels (which appear in low areas but not presenting hydrological connectivity with the sea) was automatically carried out.

Although coastal river environments might present different responses to sea-level rise than coastal areas [61], due to the proximity to the open ocean and the low-lying nature of these environments, the inundation mapping considered the same TWL values of coastal areas. Thus, the same TWL value was applied to cells with direct connectivity to coastal rivers with direct contact with the ocean (less than two bifurcations) for Suzana area, and with connectivity with the Geba river and its direct branches for Bissau study area. All other river branches were ignored as coastal flood sources.

The inundation mapping procedure did not consider possible future adaptations, such as protection structures, changes in land use and/or natural adaptation of coastal morphology to change in sea-level over time.

*3.3. Land Cover*

To assess the type of land cover affected by coastal flooding scenarios, the land cover information was crossed with the inundation extent for different SLR scenarios at each study site. The land cover map used was provided by Copernicus Global Land Services, is referenced to the year 2019 and have the following global characteristics: (1) discrete classification in 23 classes (six types of forests, six types open forests, shrubs, herbaceous vegetation, herbaceous wetland, moss and lichen, bare/sparse vegetation, cultivated and managed vegetation (cropland), urban/built-up, snow and ice, missing data and open sea; (2) approximately 100 m of spatial resolution at the equator, (3) the products is provided in a regular latitude/longitude grid (EPSG:4326) with the ellipsoid WGS 1984 [62].

The Guinea-Bissau study areas only presented the following land cover classes: (a) Bare/sparse vegetation; (b) Cultivated and managed vegetation/agriculture (cropland); (c) Urban/built up; (d) Forest (which includes the classes (i) closed forest, deciduous broad leaf, (ii) closed forest evergreen, broad leaf, (iii) closed forest, unknown, (iv) open forest, deciduous broad leaf; (v) open forest, evergreen broad leaf and (vi) open forest, unknown); (e) Herbaceous (which includes the classes herbaceous vegetation and herbaceous wetland); (f) Shrubs; and (g) Water bodies (which includes classes open sea and permanent water bodies).

## 4. Results

The projection of the TWL on the DEM allowed to evaluate the area at risk of coastal flooding in each study area. The area potentially affected by the flood in each location for different periods considered is shown in Table 4.

**Table 4.** Total flooded area in different periods in each study area, Bissau, Bubaque, and Suzana.

| Year | Bissau $km^2$ (%) | Bubaque $km^2$ (%) | Suzana $km^2$ (%) |
|---|---|---|---|
| 2041 | 12 (13) | 7 (11) | 107 (28) |
| 2083 | 25 (27) | 10 (17) | 163 (42) |
| 2100 | 27 (29) | 13 (21) | 200 (52) |
| 2100 * | 34 (37) | 20 (34) | 256 (67) |

* Extreme scenario, considering potential collapse of West Antarctic Ice Sheet.

Results show an increase in the affected area with each time horizon/SLR scenario for each considered area. In order of increase: Bubaque is expected to lose 11%, 17%, 21%, and 34%; Bissau 13%, 27%, 29%, and 37%; and Susana 28%, 42%, 52%, and 67%, respectively, by 2041, 2083, 2100, and 2100 considering the potential collapse of West Antarctic Ice Sheet. Suzana section has the largest affected area in terms of percentage in relation to the total area in all SLR scenarios. The exposed area is also affected at different paces: locations, the flooded area increases with the time horizon/SLR. For example, comparing the year 2041 with the year 2100, there is an increase in the flooded area of 11%, 16%, and 24% for Bubaque, Bissau, and Suzana, respectively. Considering the extreme SLR scenario (5 m) for the same period, the area at risk of flooding increases considerably: 23% for Bissau and Bubaque, and 39% for Suzana.

The results of the extent susceptible to coastal flooding for different areas and periods, superimposed on the satellite image, is illustrated in Figures 4–6. The visualization of the flood map allows to perceive spatially the distribution of the exposure and possible impacts of different SRL scenarios. In regard to the spatial pattern of the mapped inundation, the existence of river arms connected to the sea is paramount to allow the inundation areas to be represented further inland. This is particularly visible in Bissau, where a big low-lying area is inundated further in land, in contrast with Bubaque, where inundation patterns are all represented along the out-skirts contours of the island and never in-land.

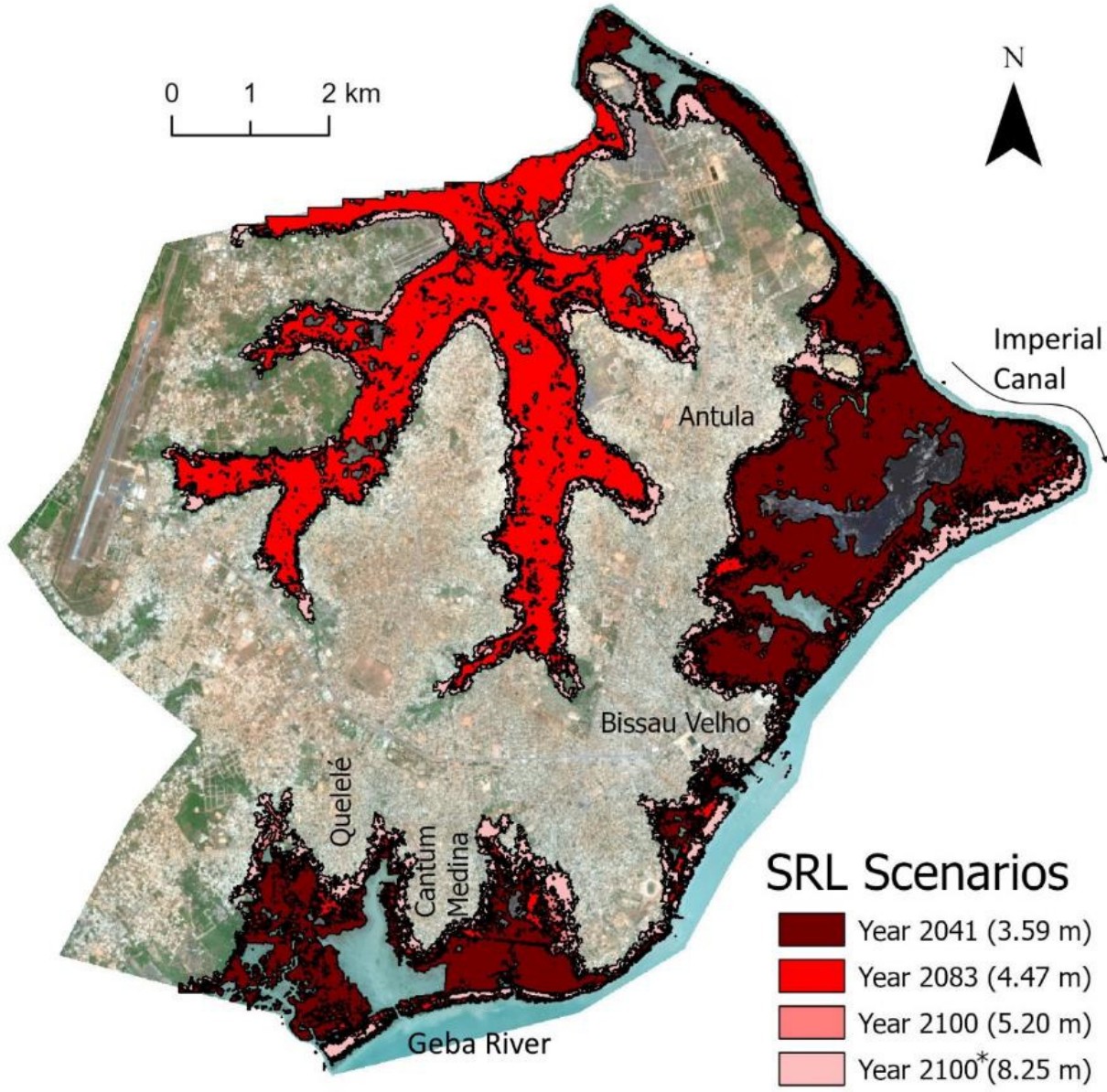

**Figure 4.** Model of the flood extent at Bissau city for different SRL scenarios. * is the extreme scenario, considering potential collapse of West Antarctic Ice Sheet. Basemap is a Sentinel 2 image.

Crossing inundation extent and land cover data allowed to estimate the area of land cover types affected by coastal flooding. The land cover types affected by coastal flooding for each study area in different SLR scenarios is shown in Figure 7. In all SLR scenarios, the land cover type most affected by flooding in Bissau is agricultural land, while in Bubaque the forest represents the most affected area and in Suzana, is herbaceous type (see Figure 7). Looking at the urban/built up area, Bissau has a much larger affected area than the other study areas, also the affected area increases as a larger SLR scenario is considered.

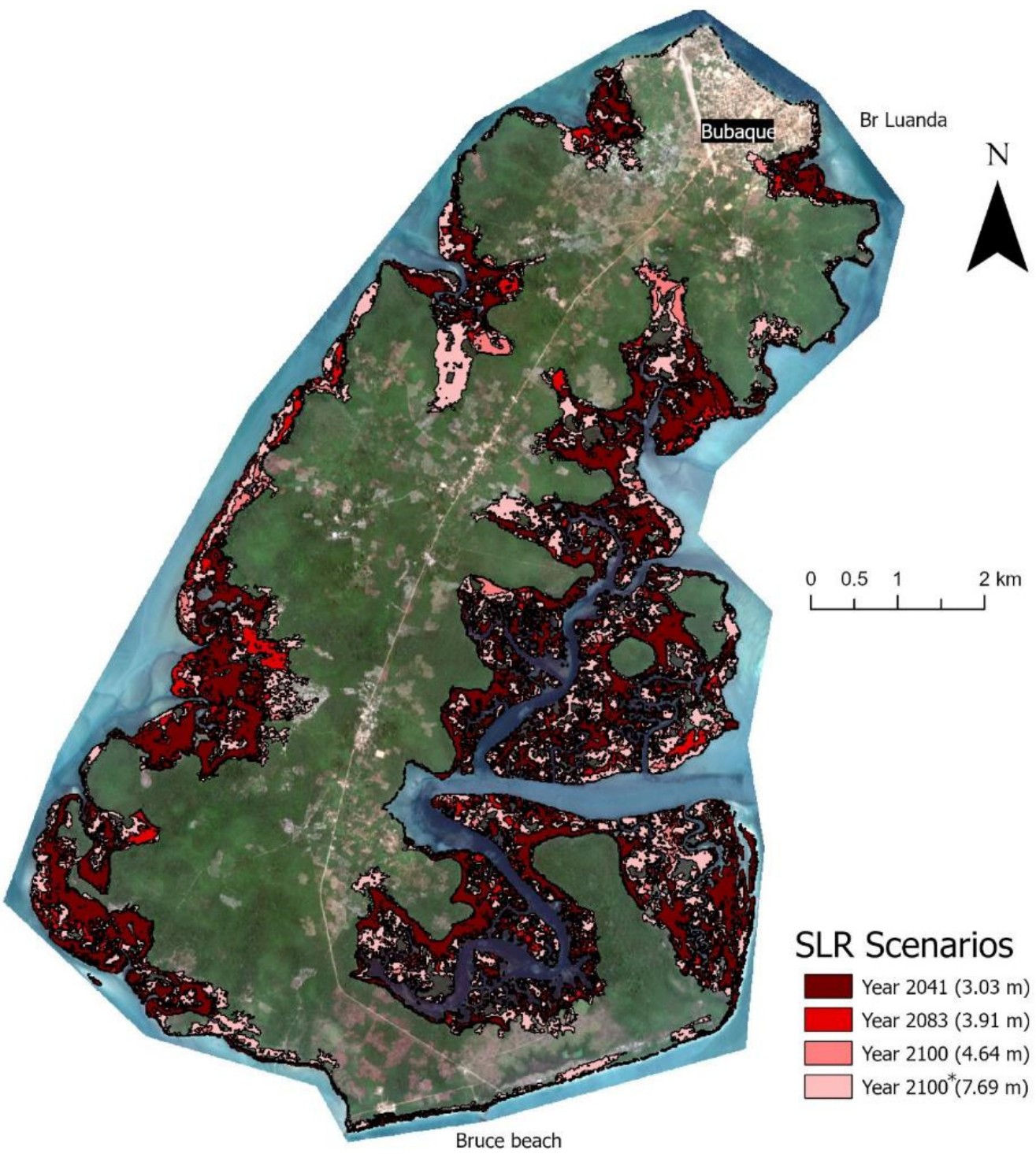

**Figure 5.** Model of the flood extent at Bubaque island for different SRL scenarios. * is the extreme scenario, considering the potential collapse of West Antarctic Ice Sheet. Basemap is a Sentinel 2 image.

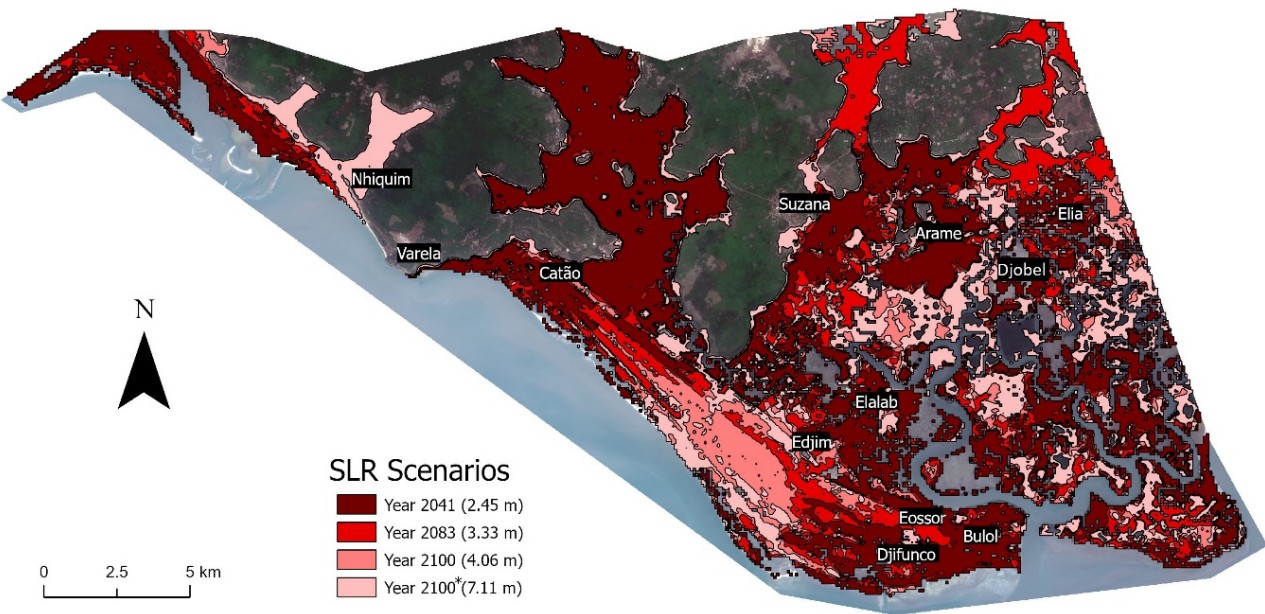

**Figure 6.** Model of the flood extent at Suzana section for different SRL scenarios. * is the extreme scenario, considering the potential collapse of West Antarctic Ice Sheet. Basemap is a Sentinel 2 image.

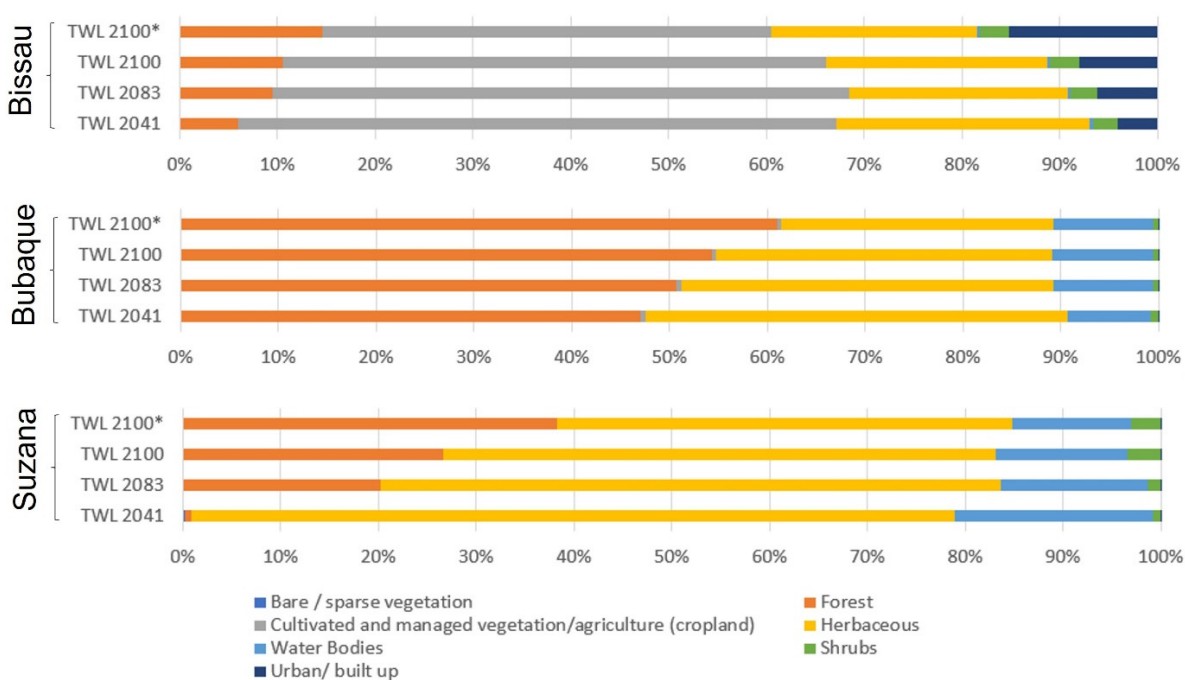

**Figure 7.** Land cover types for each study area that will be affected by coastal flooding in different SLR scenarios. * is the extreme scenario, considering the potential collapse of West Antarctic Ice Sheet.

## 5. Discussion

This study presents a simplified approach, the single surface model, coupled with high resolution global Digital Elevation Models, to assess the exposure to coastal flooding hazard in a least developed country (LDC)—Guinea-Bissau. The approach presents the mapping of coastal flooding exposure to sea-level rise under four different scenarios and at three hotspots of coastal hazards: Bissau, Bubaque, and Suzana.

The methodology here presented used the TWL approach to estimate extreme sea-level rise scenarios for 2042, 2083, and 2100, with and without the potential collapse of West Antarctic Ice Sheet. The TWL estimated in this work are in agreement with estimations

made by [63] where, for the area of Guinea-Bissau, extreme sea-levels between 5 and 10 m with a return period of 100 years are presented.

The approach did not consider the contribution of inland flooding (rainfall). Additionally, the TWLs considered only storm surge levels similar to the ones already reported for the area, and no changes or increase in storminess, which would affect the frequency of extreme events and possibly even increase the storm surge effects [64]. Furthermore, this approach did not consider the natural adaptation of the coastal area to the increase in sea-level rise, by retreating of the coast but maintaining its natural configuration. Instead, the model was applied as if the sea-level rise was instantaneously established on the coastal system, with no time for adjustments. However, the coastal inundation simulation used hydrological-connectivity with the source of flooding—the sea, which is more appropriate as directly influences only the adjacent land zones [65,66]. The authors are aware of the limitations that this approach presents to the estimation of the natural reconfiguration of this type of coastal systems but reiterate that this is a first approximation to the problem. The methodology presented here is similar to the one used by NOOA [67] to support U.S.A. coastal communities by communicating flood exposure and potential impacts though maps. Additionally, is a powerful tool for the education and awareness of the populations and competent authorities [30], thus enabling decision-makers and coastal communities to adopt appropriate adaptation measures.

Another important aspect of the presented methodology is the application of the inundation model using a high-resolution global DEM. The use of the TanDEM-X dataset allowed to better represent the coastal area inundation, instead of using other freely available global DEMs (e.g., SRTM DEM). For example, the SRTM DEM dataset could not be used because these coastal areas presented large elevation errors. As an example, SRTM DEM hindered the mapping of inundation extent for the lowest sea-level rise scenarios, because the estimated TWLs were not represented on the DEM backshore area. The overestimation of elevations in low-lying areas by the SRTM DEM was already reported many studies [66,68–70]. As concerns the use of TanDEM-X in comparison to the MERIT DEM dataset, the misrepresentation of lower elevation values was not a problem. Nevertheless, TanDEM-X presented a larger spatial resolution than MERIT DEM, which is an advantage in this type of studies. A thorough analysis on the differences of application between different DEM datasets should be conducted for coastal areas as Guinea-Bissau, similarly to the work of [16], but a high-resolution–high-precision DEM is necessary, which is not yet available for these study areas. The comparison between MERIT DEM and TanDEM-X at 90 m suggests that, for TanDEM-X_90, higher ground areas presented greater elevation values. The opposite behavior is shown for lower areas. TanDEM-X_90 presents lower elevation values for low-lying regions. The reason for this is not clear but might be related with the presence of vegetation. As discussed by [16], mangrove areas and large patches of tall vegetation present larger elevation values for TanDEM-X than MERIT DEM, when compared with high-precision LiDAR data. Nevertheless, what is clear from the application presented in this study is that global DEMs are an important tool to use for remote or poorly mapped areas, thus allowing to perform coastal inundation studies for areas with previous inexistent data.

*Implications for the Study Area*

The exposed area results show that the extent of flooding varies along the coast and in the different SLR scenarios considered, which indicates the spatio-temporal variation of the susceptibility to flooding. Additionally, note that the differences observed in the rate of increase in the flooded land are strictly associated with the topography, coastal slope and hydrological connectivity with the sea of the coastal lowlands in relation to the considered SLR scenario. Another important element that reflects in the quantity of flooded land is the variation in the TWL, which is due to differences in AT in the three locations. Additionally, there is a larger uncertainty associated with the flooded area for the Suzana in relation to the other areas in the study, since the DEM used has lower resolution/precision.

Crossing the inundation extent with the land cover data allowed to better understand the land cover types affected by coastal flooding, and, thus, perceive the influence of this factor in the exposure to this hazard. The high population density and disorderly occupation of low-lying lands in Bissau play a key role in exposure to flooding.

In Bissau, the flooding penetrates to the inland through the Geba River and Imperial Canal, covering low wetland/swampy zones, that meander the city, reaching several kilometers (see Figure 4). These lowlands, used mainly for rice growing and horticultural production, are, in general, already areas naturally flooded, but converted into cultivation fields, locally known as *bolanha* (large swampy terrain, usually near a river, where rice is grown or can be grown), through the construction of "anti-salt" dikes located downstream of the river arms, thus preventing the passage of salt water upstream. Presently, rice crops in these areas are sometimes affected by coastal flooding when, besides the overwash, there are also the breaking of anti-salt dykes due to high tides or SS, as happened when Tropical Storm Fred in 2015 hit Bissau. SLR will tend to reduce the urban and peri-urban agricultural land, as ever-larger areas will become permanently or frequently flooded, compromising the cultivation of rice and other crops, livelihood of many families, which, according to [71], are already threatened by the rapid disordered urbanization in recent decades.

Additionally, the advance of the urban net to areas with low-lying land susceptible to coastal flooding will put increasingly more people, housing, and infrastructure at risk as sea-level rises. The southwest zone of the city, located in low-lying areas, which is predominantly residential, and with an extensive built area and a high number of houses in precarious conditions, will be the most affected, particularly the Cuntum Madina and Quelelé neighborhoods (see Figure 4). It should be noted that these neighborhoods are already experiencing severe coastal flooding in areas with lowest elevations, mainly during spring tides. In recent years, due to the recurrent coastal flooding in these areas, some residents were forced to abandon their homes, while others continue to live with the risk, which also threatens public infrastructures, such as schools (Figure 8A–C). The old part of the city, known as "Bissau Velho", whose part has developed on low-lying land and where offices and commercial activity are concentrated, is also already affected by coastal flooding. For example, the barracks of the National War Navy, located in this area, are already periodically flooded by spring tides (Figure 8D), with the General Staff of the Armed Forces and Government planning to move the military to safer facilities. An example of other zones, where urbanization is advancing on floodable land, is the low-lying wetland located between the neighborhoods of "Bissau Velho" and the Antula (see Figure 4), where, in recent years, there was an explosion of construction of enormous storehouses and factories. According to Fandé et al. [28] the increasing advance of urbanization to floodable areas is related to the rapid population growth of the city, lack of awareness of the population, and weak intervention of the municipal authorities and the central government.

If the SLR scenarios considered in this study are verified and if adaptation measures fail to be adopted, the risk of flooding will tend to worsen progressively in the aforementioned zones, affecting others currently unaffected areas (e.g., port terminals located in Bissau Velho), reaching a particularly serious situation in the extreme scenario of 2010 (see Figure 4). The most inland zones of the city will be affected to a lesser extent, due to their relatively higher topography and/or the existence of more obstacles, such as dikes and roads.

In Bubaque the flooding extends mostly to rural areas and away from settlements, affecting forest and herbaceous areas, especially mangrove areas and, to some extent, areas of palm trees and other arboreal vegetation. In the Bubaque city, where most of the island's population and infrastructure are concentrated, the flood extends more to the lower elevation zones located to the southeast (Luanda Neighborhood), than in other zones with higher elevations (see Figure 5). At Bruce beach, in the south of the island, where some hotels are presently affected by spring tides, there should occur damages to these type of structures by 2041, due to permanent and/or more frequent flooding. Coastal flood should reach relatively higher zones at Bubaque city from 2083, with a greater extent if the collapse of Antarctica occurs in 2100 (see Figure 5), which would cause damage to infrastructure

very close to the sea, namely tourist infrastructure and the port terminal. It should be noted that in the Bubaque island the agriculture is not practiced in *bolanha*, usually the Bijagó (the largest ethnic group on the island) practices rainfed agriculture on the islands near Bubaque, mainly on the Rubane island.

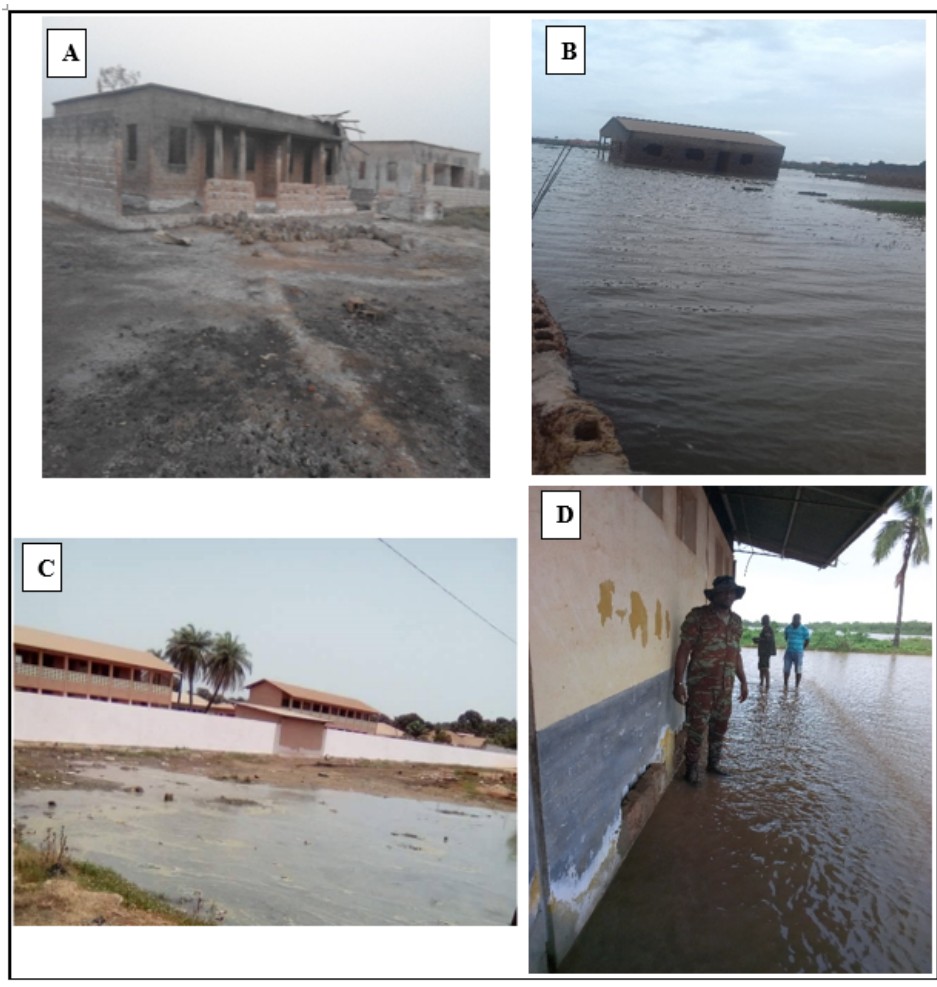

**Figure 8.** Effects of coastal flooding in Bissau: (**A**,**B**)—houses abandoned due to coastal flooding in the Cuntum Madina neighborhood; (**C**)—school in a coastal flood risk area in the Cuntum Madina neighborhood; (**D**)—flooding by spring tides in the Navy headquarter in Bissau Velho.

In Suzana, the flooding extends more to southeast, precisely close by the Cacheu River estuary, a low-lying region where several villages have developed, some of them completely isolated by river arms: Djobel, Elia, Bulol, Djifunco, Eossor, Elalab, Arame, and Edjim (see Figure 6). These villages, which concentrate a large part of the Suzana's population [36], are already affected by flooding, which mainly impacts agricultural land and housing. Of these villages, Djobel, an island with 223 inhabitants [36], deserves special attention. The village is practically covered by sea water during spring tides, forcing the inhabitants to build dikes and to raise the land around the houses to protect them (Figure 9). In the period of spring tides, the mobility is reduced since, at that time, the movements from one *morança* (Set of houses, in a village, in which inhabits a single familiar aggregate) to another can only be done by canoe or using the dike pathway, which have an insufficient coverage. Agriculture on this island has been affected by saltwater intrusion in cultivation fields, due to the frequent rupture/overtopping of anti-salt dikes by the sea water. Additionally, due to saline intrusion, the inhabitants of Djobel no longer have fresh water, having to travel some miles by canoe to fetch it from neighboring villages.

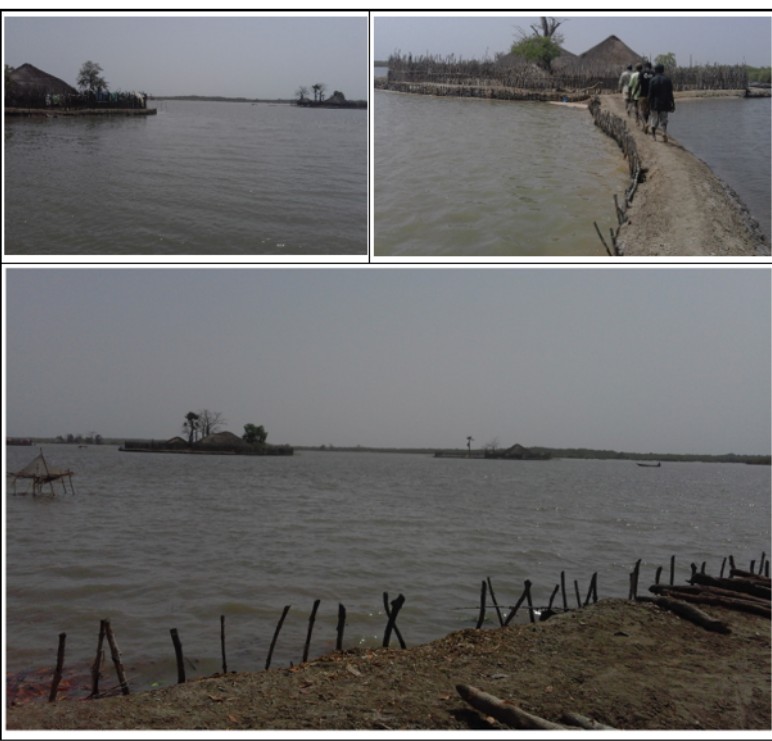

**Figure 9.** Flooding by spring tide, also showing the elevation of land around the houses in the village of Djobel (Suzana).

The flooding manifests with greater intensity during extreme events, such as storms, and its impacts become more devastating. For example, in August 2015, the high storm surge induced by Tropical Storm Fred flooded intensely Djobel and Elalab, destroying dikes and leaving enormous crop losses, as well as damage to housing, among others. At that time, the committee of Djobel indicated that "the salt water has spoiled all the *bolanhas*, entered in our homes and at this moment everyone, including women, is obliged to practice fishing activities in order to survive" [72] (p. 10).

The situation of the inhabitants of Djobel has worsened and the authorities are already taking steps to relocate them to the mainland. However, there has been resistance from traditional occupants of the new area used to resettle the victims, which has recently resulted in confrontations. In this regard, C. J. Insumbo indicated that there was a first confrontation with shots, in February 2019, between the inhabitants of Djobel and Arame, caused three injured, in the wake of the inhabitants of Djobel having carried out deforestation and cutting of cashew trees on the mainland part, on the border between Elia and Arame, which had been made available to them by the administrative authorities of the São Domingos Sector (personal email, 4 June 2019). He also said that on 31 May 2019, a delegation from the Ministry of Interior went to the conflict zone and together with the Association of Sons and Friends of the Suzana Section (AOFASS), brought together the conflicting parties, and the authorities guaranteed to find a solution that would allow to relocate the population of Djobel to the mainland as soon as possible, through participatory processes to be triggered by the Ministry of Territorial Administration (personal email, 4 June 2019).

If the SLR scenarios considered in this study are verified and if adaptation measures are not taken, the villages of Djobel, Elabab, Elia, Bulol, Eossor, Djifunco, Arame, and Catão will need to be abandoned before the year 2041; and by 2083 the village of Edjim will go through the same situation as currently Djobel is facing and will need to be abandoned as well; and if the collapse of the Antarctica occurs in 2100, Suzana and Nhiquim will also be affected (see Figure 6).

The *felupe*, main ethnic group that inhabits Suzana, traditionally lives near the sea/river, where they practice a subsistence agriculture (basically rice cultivation in flooded lands)

and fishing as main activities. If the projected SLR scenarios are observed, the risk of flooding will be increased and could seriously compromise the level of subsistence of many families and contribute strongly to the increase in rural exodus and climate refugees. Additionally, as emphasized by [73], accommodate these migrants, forced to leave their lands, would create a major social and political problem, such as internal conflicts, poverty, and diseases, for example.

It should also be noted that at Suzana, the villages are small and have developed with the preservation of the natural environment, which is reflected in the little representation of the built-up area affected by flooding, and a greater representation of affected areas covering different classes of vegetation (see Figure 7).

## 6. Conclusions

This study used a simplified methodology, which considers a single-value model, to assess flood risk in three coastal areas of Guinea-Bissau, in three period with different SLR scenarios (2041, 2083, and 2100), together with global digital elevation models. The methodology, although simple, is able to provide a first snapshot of the inundation extent configuration on different sea-level rise scenarios. Additionally, the use of global DEMs can aid on the application of coastal inundation modelling in areas where no other detailed datasets are available.

The land area potentially at risk of flooding increased significantly and progressively from 2041 to 2100 in all of study areas. If the future SLR scenarios considered in this study occur, it could be seen, in the coming decades, an increasing economic and social disruption in Suzana area because of loss of property and livelihoods and forced migration. Bissau also will face serious problems, such as loss of infrastructure and agricultural land; while Bubaque will be less affected. It can be considered that coastal flooding will be one of the most significant impacts of climate change in the three study areas during the 21st century, and this work provides the first steps for increasing awareness of these types of climate change associated risks. As can be expected, adaptation measures to mitigate climate change will be a great challenge for homeowners, farmers, and authorities all around the world, but specially in least developed countries, as they have to conciliate risk and poorer conditions to tackle that risk.

The methodology here applied can be easily used to carry out similar studies in other coastal locations, where detailed information on the variables at play are difficult to obtain or nonexistent. The results presented can be used in a practical way in adaptation scenarios, as they provide researchers and policy makers in least development countries a view of the potential impacts of coastal flooding in the present and future, thus aiding on the conception and implementation of sustainable adaptation policies. The consideration of future SLR scenarios in the planning of use and occupation of urban/rural land can avoid the inappropriate occupation of areas at risk of coastal flooding, directing urbanization and economic activities to the safest ones, making them more adapted and resilient to possible climate changes.

**Author Contributions:** Conceptualization, Morto Baiém Fandé, Cristina Ponte Lira and Gil Penha-Lopes; methodology, Morto Baiém Fandé and Cristina Ponte Lira; writing—original draft preparation, Morto Baiém Fandé, Cristina Ponte Lira and Gil Penha-Lopes. All authors have read and agreed to the published version of the manuscript.

**Funding:** This research was funded through the provision of high-resolution TanDEM-X dataset by © DLR 2017, under the framework of project DEM_OTHER0576.

**Institutional Review Board Statement:** Not applicable.

**Informed Consent Statement:** Not applicable.

**Data Availability Statement:** Data is protected under undisclosed agreements with © DLR.

**Acknowledgments:** The authors would like to thank the Fundação para a Ciência e a Tecnologia (FCT) for supporting the Ph.D. PD/BD/114055/2015 of Morto Baiém Fandé, Cristina Ponte Lira thanks the FCT project UIDB/50019/2020—IDL—Instituto Dom Luiz, the researcher contract (IF/00940/2015) of Gil Penha Lopes, the research center grant UIDB/00329/2020—Centre for Ecology, Evolution and Environmental Change (cE3c). Special thanks to: German Aerospace Center (DLR), for providing the DEM: Elevation data is a TanDEM-X Digital Elevation Model, derived from TANDEM-X mission and provided by ©DLR 2017 in the scope of project DEM_OTHER0576. A special thanks to Professor Carlos Antunes in helping with tidal data for Guinea-Bissau.

**Conflicts of Interest:** The authors declare no conflict of interest.

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
