# Peer review of "Using TanDEM-X Global DEM to Map Coastal Flooding Exposure under Sea-Level Rise: Application to Guinea-Bissau"

_ijgi, doi:10.3390/ijgi11040225_

Round 1

Reviewer 1 Report

Dear Editor;

I have the following comments before acceptance of the mentioned manuscript.

 Abstract

Any quantitative results in abstract?

Keywords: please use upper case in this word (bathtub model)

  1. Introduction

What is innovative of the current study?

  1. Study Areas

Please add coordinate system of the mentioned studied areas.

  1. Materials and Methods

Please add a flowchart for this methodology.

Figure 2.

I think this figure is your result. Isn’t a method.

  1. Results and Discussion

Please write these two parts separately. This is important to add some comparisons to previous publications and explain.

Reviewer 2 Report

The manuscript titled "Using TanDEM-X global DEM to map coastal flooding expo-2 sure under sea level rise: application to Guinea-Bissau" aims to map coastal areas exposed to flooding hazard under 2041, 2083 and 2100 sea level rise scenarios. The used method applies the simple “bathtub” model coupled with high-precision digital elevation models. In its simplicity it is substantially simple and trite, and does not require changes other than a careful prooforeding of the whole text. However, I suggest to evaluate if the term "hazard" instead of "risk" could be more appropriate.
There are some terminological errors (as an example "exposer" in place of "exposure", line 13 of the Abstract). Other lines are not understandable ("The area potentially exposed to coastal flooding was evaluated in a geographic information systems (GIS) environment. together different and discussed other elements at risk in. Terrain elevation data were obtained from a. was used to assess the potential flood risk trough", lines 21-24). Again, some repetitions are also present (e.g. "concluding that the average vertical accuracy of TanDEM-X 90 and MERIT are similar and the average vertical accuracy of TanDEM-X 90 and MERIT are similar, being both a significant improvement on SRTM." lines 73-75). Finally, there are also some sentences of the template ("This section may be divided by subheadings. It should provide a concise and precise description of the experimental results, their interpretation, as well as the experimental conclusions that can be drawn", lines 521-523).
However, the effort made by authors to map such an above hazard is noticeable taking into account the absence of detailed topographic map and accurate elevation data for the studied areas.

Reviewer 3 Report

The manuscript is generally well organized and well written.  There are several minor English-usage errors, listed in detail below, but certainly not an impediment to understanding. The science is clear, and the conclusions appear fine.  The use of the higher resolution data  elevates this above some similar documents I have read.  I believe everyone understands that the bathtub model is only a first approximation, particularly on the spits, but it is probably the best that can be achieved in this less-well-developed setting.  The results are definitely local, but should be of interest to a broad audience, and certainly have important significance to the inhabitants of these coastal systems.

References, Figures and Tables are all good.  I believe Table 1 would be improved by changing the text entries to right and left justified individual columns, rather than the existing centered form.  This would improve readability.

LINE #

003     sea-level rise: hyphenate compound modifiers. Check here and throughout.

013     Sp. “exposure”

014     sea-level rise: see note 003

023     “in.” – something is missing

024     “a.” – undefined

028     REWRIE to: “and a rain-fed agriculture is practiced,”

030     replace “low” with “less” or “least”

034     sea-level rise

046     “infrastructure”   (composite noun written in singular form)

059     “non-freely available version”

066     space after “than”

068     space after “%”

069     “provide” plural for multiple-author paper

080     remove “a” – plural ecosystems and wetlands

081     use superscript form for km2

082     “Guinea-Bissau’s” possessive form

090     “give” – see note 069

093     “at the local level”

103     “more likely”

114     Sp. “and”

123     “activities of “

133     “allows representation of a”

143     “sea-water levels” see note 003

150     It is generally expected that maps have latitude and longitude marked, in this case at least for the intermediate scale map of the west African coast

156     “close to” instead of “closed to”

170     “Tropical Storm Fred”

177     Table 1 would be more easily readable if the columns with text were given specific margins left and right, rather than centered. The numeric values work okay as is (centered).

181     “Sea-level rise”

189     “continental shelves are”

192     “CO2” use subscript form

199     “the West Antarctic Ice Sheet”

219     “the West Antarctic Ice Sheet”

239-240         space before “m”

257     “e.g.,” proper punctuation always includes the comma, before a dependant phrase.

282-283         space before “m” – please be consistent

292     REWRITE “a risk of flooding was considered for any”

312     “has” singular, referring to “map”

316     space before “m”

344     “West Antarctic Ice Sheet”

372     “Tropical Storm Fred” remove unnecessary “the”

401     “of the West Antarctic Ice Sheet”

413     “e.g.,”

446     “dyke” or “dike” – be consistent throughout

457     unnecessary “the”

521-523         Text inadvertently left over from the formatting document

647     “CO2” use subscript form

Reviewer 4 Report

The article is interesting, and the subject is worthy of research. However, the execution of the article and the research as well as the presentation itself requires important improvements to proceed with its publication in the journal in my opinion. For this I advise a major revision to the authors in the following points:

Several issues on different figures need to be improved. Some of them are missing the scale bar. Additionally, no source of background map is presented. Are all elements on those maps are your own production? If not a reference is needed.

In the introduction, the authors miss a number of recent articles that deal with coastal flooding exposure and flood risk assessment suggest that the literature review is still to be done. Please add more positions concerning floods in general and some examples of works dealing with different types of floods. Consider works like Salman, AM (2018), Zischg, AP (2018), Dandapat, K (2017), Paprotny, D (2018) among others. Finally, it is necessary to especially present more examples of similar works that deal with the “bathtub” model like Paprotny D (2017), Williams LL (2020), Didier D (2019).

It is unclear to me what is actually the main aim of the work and also what is the significance and innovation of the work discussed in the paper, compared to other similar works. As authors state themselves (lines 30-31) the methodology presented in the manuscript is simple and also well-known it is necessary to underline the novelty of the paper.

Chapter 4  “Results and discussion” is more like results only. There is no discussion at all. I suggest to intensively go through the literature and rewrite the discussion part. Additionally, the results themselves are very poorly presented and they reduce the whole work just down to a simple figure description. Authors need to extend this part and present more details of achieved results.

What about climate change? Please keep in mind that climate changes will not only lead to precipitation increase or sea level rise, but may also result in changes in storminess, which affects the frequency of extreme events. In this case it should be at least mentioned about some possibilities of adaptation measures.

The Conclusion does not illustrate the article. According to Introduction author states that “This study aimed at quantifying and map the exposed area to the risk coastal flooding using SLR 18 scenarios…” while conclusion underline the methodology usage neglecting the main aspect of the manuscript underlined in Introduction.

Lines 521-523 seems to be some kind of instruction for result section that was not deleted from the MDPI template.

Round 2

Reviewer 4 Report

Congratulation to the authors.

I accept implemented changes.